# Incidence and influencing factors of occupational pneumoconiosis: a systematic review and meta-analysis

Xuesen Su [ID],[1,2] Xiaomei Kong,[2,3] Xiao Yu,[2,3] Xinri Zhang[2,3]

[1]The First College for Clinical Medicine, Shanxi Medical University, Taiyuan, Shanxi, China
[2]The National Health Commission Key Laboratory of Pneumoconiosis (Shanxi, China) Project, First Hospital of Shanxi Medical University, Taiyuan, Shanxi, China
[3]Department of Pulmonary and Critical Care Medicine, First Hospital of Shanxi Medical University, Taiyuan, Shanxi, China

**Correspondence to**
Mr Xinri Zhang;
XinriZhangsxmd@163.com

## ABSTRACT

**Objectives** To determine the incidence of pneumoconiosis worldwide and its influencing factors.

**Design** Systematic review and meta-analysis.

**Setting** Cohort studies on occupational pneumoconiosis.

**Participants** PubMed, Embase, the Cochrane Library and Web of Science were searched until November 2021. Studies were selected for meta-analysis if they involved at least one variable investigated as an influencing factor for the incidence of pneumoconiosis and reported either the parameters and 95% CIs of the risk fit to the data, or sufficient information to allow for the calculation of those values.

**Primary outcome measures** The pooled incidence of pneumoconiosis and risk ratio (RR) and 95% CIs of influencing factors.

**Results** Our meta-analysis included 19 studies with a total of 335 424 participants, of whom 29 972 developed pneumoconiosis. The pooled incidence of pneumoconiosis was 0.093 (95% CI 0.085 to 0.135). We identified the following influencing factors: (1) male (RR 3.74; 95% CI 1.31 to 10.64; p=0.01), (2) smoking (RR 1.80; 95% CI 1.34 to 2.43; p=0.0001), (3) tunnelling category (RR 4.75; 95% CI 1.96 to 11.53; p<0.0001), (4) helping category (RR 0.07; 95% CI 0.13 to 0.16; p<0.0001), (5) age (the highest incidence occurs between the ages of 50 and 60), (6) duration of dust exposure (RR 4.59, 95% CI 2.41 to 8.74, p<0.01) and (7) cumulative total dust exposure (CTD) (RR 34.14, 95% CI 17.50 to 66.63, p<0.01). A dose–response analysis revealed a significant positive linear dose–response association between the risk of pneumoconiosis and duration of exposure and CTD (P-non-linearity=0.10, P-non-linearity=0.16; respectively). The Pearson correlation analysis revealed that silicosis incidence was highly correlated with cumulative silica exposure (r=0.794, p<0.001).

**Conclusion** The incidence of pneumoconiosis in occupational workers was 0.093 and seven factors were found to be associated with the incidence, providing some insight into the prevention of pneumoconiosis.

**PROSPERO registration number** CRD42022323233.

## STRENGTHS AND LIMITATIONS OF THIS STUDY

⇒ To our knowledge, this is the first systematic review and meta-analysis to estimate the incidence of pneumoconiosis in dust-exposed workers by pooling data from only cohort studies.

⇒ Our study provided quantitative relationship between influencing factors and pneumoconiosis with a larger sample size.

⇒ Due to the limited literature available, some potentially relevant risk factors, such as alcohol consumption and tuberculosis, were excluded from this meta-analysis.

⇒ Significant heterogeneity in the pooled results for a variety of reasons and the small number of included studies may limit the dependability of our results for certain risk factors.

lung dysfunction. Although pneumoconiosis among dust-exposed workers has received considerable worldwide attention in terms of incidence, risk factors, early diagnosis, prevention and intervention, there are currently few established therapeutic options for the disease, highlighting the importance of precautionary measures.[1]

Methods of assessing exposure, controlling dust concentration, using tools to prevent exposure, undergoing regular physical examination and preserving pathological records have been implemented to protect workers from dust inhalation, and eventually, some studies have reported a downward trend since 2015[2–5]—around 527 500 cases of prevalence with about 60 000 new patients reported globally in 2017 corresponds to around 1 111 000 cases of prevalence with about 99 000 new patients in 2016. However, it is still escalating in certain regions, particularly in countries with a large labour force.[6 7] In addition, there are some worse outcomes for a subset of employees. In a 14-year cohort study, Poinen-Rughooputh *et al*[8] found that the incidence and cumulative risk of silicosis increased more rapidly with increasing years of smoking and dust exposure compared with

## INTRODUCTION

Pneumoconiosis is inclusive of a group of serious pulmonary diseases associated with the inhalation of mineral dust and corresponding reactions of lung tissues, including diffuse fibrosis and progressive

never-smoking workers, indicating that stricter exposure limits and additional preventative measures may need to be imposed on smoking workers. Therefore, the possible reason for the current situation is that current preventive standards may not be able to protect workers with vastly different characteristics, in other words, targeted prevention may be in high demand at this time.

Targeted prevention requires a deeper comprehension of the influencing factors. Even though previous studies have demonstrated that the incidence rate of pneumoconiosis is affected by a variety of factors, including advanced age, male gender, smoking and dust exposure,[8–10] limited prospective data and the homogeneity of studied populations impede their clinical applicability. Furthermore, there is no meta-analysis or systematic review that summarises the relevant research findings. Consequently, pneumoconiosis risk factors and the shape of their relationship remain largely unknown.

We conducted this systematic review and meta-analysis to provide some clues for the targeted prevention of pneumoconiosis in light of the paucity and controversy of evidence currently available. With a larger sample size, this meta-analysis sought to address and explore the incidence and the impact of different influencing factors on the morbidity of pneumoconiosis, as well as the shape of the relation between them.

## METHODS
This meta-analysis was completed in accordance with the Preferred Reporting Items for Systematic Reviews and Meta-Analyses statement[11] and was registered with PROSPERO (Registration NO: CRD42022323233).

### Search strategy
By using search terms related to "pneumoconiosis" and "cohort studies", two reviewers independently and systematically searched through online databases, including PubMed, EMBASE, the Cochrane Library and Web of Science, before November 2021, without regard to the language of publication. Through a thorough discussion with a third reviewer, any discrepancies in the literature search process were resolved. Each electronic database's search terms are detailed in online supplemental table 1.

### Inclusion and exclusion criteria
Two reviewers exported all studies identified by the search strategy to Endnote V.X9 independently (Thomson Reuters, Philadelphia, USA). The titles and abstracts of all search results were screened first, followed by an independent examination of the full texts of all eligible literature. This meta-analysis required the following inclusion criteria for studies: (1) all involving occupational dust exposure workers (exposed to occupational dust in any period of their work history) are divided into with-pneumoconiosis group and without-pneumoconiosis group according to the national diagnostic criteria of pneumoconiosis published by the International Labour

Organization (1980, 2000 and 2009 editions) or Ministry of Health of the People's Republic of China (GB 5906–1986, GB 5906–1997, GBZ 70–2002, GBZ 70–2009 and GBZ 70–2015)[12–14]; (2) at least one variable investigated as an influencing factor for the morbidity of pneumoconiosis; (3) the outcome was the number of workers with and without pneumoconiosis under the condition of with and without exposure or with different levels of exposure and (4) cohort study designs. The following studies were excluded: (1) studies with insufficient data and (2) the diagnosis of patients with pneumoconiosis was not reported. Disputes regarding the inclusion of eligible papers were resolved through a full discussion with a third reviewer.

### Quality assessment
The quality of the included studies was assessed independently by two authors using the Newcastle-Ottawa Scale (NOS). The NOS contains eight categories relating to methodological quality and each study was given an eventual score out of a maximum of 9 points. A score of 0–6 indicated a low-quality study, whereas a score of 7–9 indicated a high-quality study. Any disagreements about quality assessment were resolved through a full discussion with a third reviewer.

### Data extraction
Two reviewers independently conducted data extraction and any disagreements about data extraction were resolved through a full discussion with a third reviewer. The following information was extracted: (1) study characteristics: first author name, publication year, country, duration of follow-up, sample size; (2) type of pneumoconiosis included; (3) participant numbers (separating those who developed pneumoconiosis and those who did not); (4) characteristics of these groups (such as gender); (5) exposure of interest (duration or dust concentrations); (6) information about the outcome of interest (incidence of pneumoconiosis) and (7) risk estimates with 95% CIs from any statistic model.

### Statistical methods
Statistical analyses were conducted by using Statistical Package for the Social Sciences (SPSS, V.25.0, IBM SPSS Statistics; IBM), Stata V.16.1 software (StataCorp) and R V.4.1.3 statistical software (Schwarzer, 2007; TUNA Team, Tsinghua University, 2022). First, to investigate the incidence of pneumoconiosis in dust-exposed workers, the pooled incidence and its 95% CIs were calculated using a random-effects model. Second, we chose the influencing factors that were mentioned in at least two studies to investigate the relationship between those and the incidence of pneumoconiosis. A dichotomous variable was reported as frequency and proportion, and the risk ratios (RRs) and their 95% CIs were summarised to assess the outcome using a random-effects model. Third, we compared the highest categories to the lowest categories and conducted dose-response meta-analyses to investigate the association

between the risk of pneumoconiosis and dust exposure including two aspects: duration of dust exposure and cumulative total dust exposure concentration (CTD). A p<0.05 was considered significant for all tests.

First, we examined the association between exposure and pneumoconiosis risk using a random-effects model to summarise RRs and 95% CIs from each study obtained by comparing the highest versus the lowest exposure categories. The dose–response analysis was then conducted using the method described by Greenland and Longnecker[15] and Orsini et al,[16] which required knowledge of the distribution of cases, non-cases, the RRs and 95% CIs for at least three quantitative exposure categories. For each exposure category, we extracted the midpoint as the dose in each category, estimated by calculating the mean of the lower and upper bound. When the highest and lowest categories were open-ended, we assumed the length of these open-ended intervals to be the same as those of the adjacent intervals.

Using a two-stage, random-effects dose-response meta-analysis, a potential non-linear association between the risk of pneumoconiosis and two aspects of dust exposure was investigated. The modelling of dust exposure and restricted cubic splines with three knots at fixed centiles of 10%, 50% and 90% of the distribution was used to conduct this meta-analysis. We calculated restricted cubic spline models by a generalised least squares trend estimation method, which takes into account the correlation within each set of reported relative risks or HRs, based on the Orsini method. In addition, we pooled study-specific estimates using the restricted maximum likelihood method in a multivariate random-effects meta-analysis. The probability of non-linearity was estimated by testing the null hypothesis under the assumption that the coefficient of the second spline is equal to zero. Using the two-stage generalised least squares trend estimation method, a linear dose–response relationship was examined between the risk of pneumoconiosis and additional 5 years of duration and 50 mg/m³-years of CTD. First, study-specific slope lines were estimated, and then, using a random-effects model, these lines were combined to obtain an overall average slope. Finally, obtained regression values of the dose–response relationship in the spline and line models at the chosen reference level (the analysis process described above is detailed in online supplemental table 1). Statistical analyses were conducted using Stata V.16.1 software.

The association between cumulative silica exposure (CSE) and the incidence of silicosis was estimated using the Pearson correlation. Due to the incidence rate of silicosis being non-normally distributed data (the higher the incidence, the smaller number of groups), the natural logarithm of the incidence rate was used as the normal distribution variable.

Both the Q statistic and $I^2$ were calculated as indicators of heterogeneity (significant heterogeneity defined as $I^2>50\%$, p<0.05). Subgroup analyses were performed to investigate possible reasons for the significant heterogeneity and address the effects of important factors. We divided the data into subgroups based on the type of pneumoconiosis (Coal Worker's Pneumoconiosis, CWP vs other types of pneumoconiosis), occupational category for CWP, and study quality (NOS<7 vs NOS≥7), and then compared across subgroups.

Eliminating individual studies one by one, a sensitivity analysis was conducted to determine the effect of each study on the pooled estimate. Using the funnel plots and Egger's tests, we evaluated the potential risk of publication bias in the included studies.

## Patient and public involvement

Patients and/or the public were not involved in the design, conduct, reporting or dissemination plans of this research.

## RESULTS
### Study selection and study characteristics

Figure 1 presents a detailed flow chart of the literature screening. In the initial search, 5293 articles were found. After excluding papers that were duplicates or did not meet the inclusion criteria, 78 full-text articles of potentially relevant studies were identified. Following a full-text review, an additional 59 articles were excluded: 44 studies with insufficient and duplicate research data, 6 cross-sectional studies, 5 reviews, 3 case–control studies and 1 case report. Finally, 19 studies were included in our meta-analysis.

Among these studies, there were six prospective studies, nine retrospective studies and four retrospective-prospective cohort studies. The range of NOS scores was 5–9, and publish years ranged from 1981 to 2021. This meta-analysis involved 335 424 participants, including 29 972 pneumoconiosis patients and 305 452 non-pneumoconiosis workers. We analysed seven factors influencing the morbidity of pneumoconiosis, including gender, smoking, coal workers' occupational categories, duration of dust exposure, age, CTD and CSE (table 1).

### Pooled results, sensitive analysis, publication bias of the incidence of pneumoconiosis

Based on the results of the random-effects method, the pooled incidence of pneumoconiosis among dust-exposed workers was 0.093 (95% CI 0.085 to 0.135), with a high-level heterogeneity between studies ($I^2$=99.93 %, p<0.0001) (figure 2, table 2). The sensitive analysis revealed that no individual studies significantly affected the pooled results (online supplemental figure 1). Egger's linear regression test (t=0.81, p=0.427) indicated that there was no significant publication bias, and the funnel plot was presented in online supplemental figure 2.

### Subgroup analysis of the incidence of pneumoconiosis and comparison results

Table 2 demonstrates the pooled incidence of all subgroups stratified by different types of pneumoconiosis,

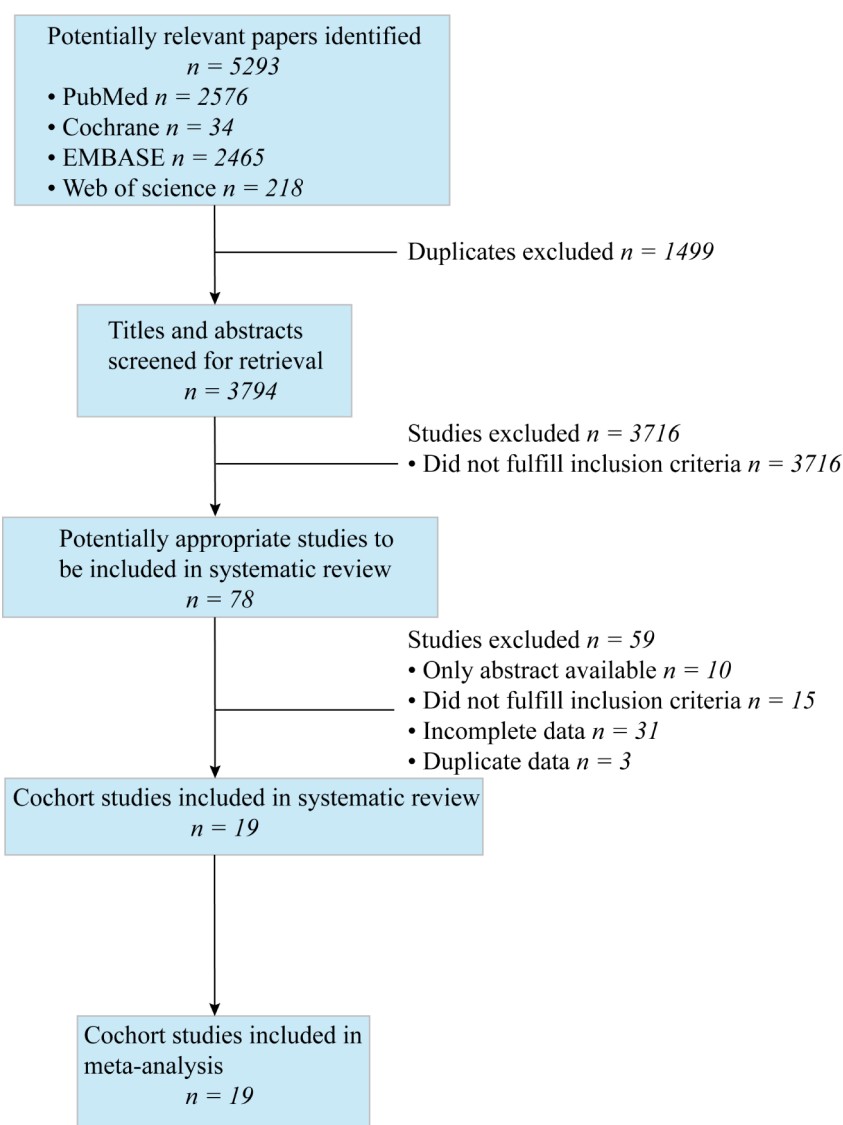

**Figure 1** Flow diagram of study selection.

gender, smoking status and occupational categories of coal workers. The pooled incidence of silicosis (0.119, 95% CI 0.064 to 0.188) was higher than that of CWP (0.044, 95% CI 0.024 to 0.082). Males had a higher pooled incidence (0.034, 95% CI 0.000 to 0.123) than females (0.012, 95% CI 0.002 to 0.086). Additionally, the pooled incidence among smokers (0.064, 95% CI 0.012 to 0.152) was higher than that among non-smokers (0.019, 95% CI 0.005 to 0.076). The pooled incidence of CWP in various occupational categories varied. Tunnelling workers had an incidence of 0.112 (95% CI 0.062 to 0.174), mining workers was 0.059 (95% CI 0.025 to 0.093), combining workers was 0.049 (95% CI 0.020 to 0.123) and helping workers was 0.005 (95% CI 0.001 to 0.019).

### Influencing factors of pneumoconiosis
We found seven factors influencing the incidence of pneumoconiosis, including gender, smoking, age, coal workers' occupational categories, duration of dust exposure, CTD and CSE.

### Gender
The association between gender and the incidence of pneumoconiosis was examined in five studies, four of which were of high quality and one of which was of low quality. Using a random-effects model, the pooled results of five studies indicated that male workers had a significantly higher risk than female (RR 3.74; 95% CI 1.31 to 10.64; p=0.01) with heterogeneity (p<0.00001, $I^2$=93%). Subgroup analysis for only high-quality studies (RR 4.54, 95% CI 3.70 to 5.57, p<0.00001) with non-significant heterogeneity (p=0.40, $I^2$=0%) revealed the possible cause of significant heterogeneity (figure 3A).

### Smoking
Five studies, all of which were high-quality studies, reported the comparison between smoking and never-smoking. The pooled results of five studies suggested that smoking workers had an obviously higher risk than non-smoking workers (RR 1.80; 95% CI 1.34 to 2.43; p=0.0001) with heterogeneity (p=0.02, $I^2$=67%) by using

**Table 1** Study characteristics

| Study | Country | Design | Pneumoconiosis reported | Duration (years) | Sample size (pneumoconiosis cases/total cases) | Risk factors reported | NOS scores |
|---|---|---|---|---|---|---|---|
| Chen et ai[9] | China | Retrospective | Silicosis | 45 | 4465/23002 | F5 | 7 |
| Morfeld et al[54] | Germany | Prospective | Silicosis | 20 | 40/17144 | F1, F2 | 8 |
| Zhang et al[17] | China | Prospective | Silicosis | 29 | 48/2009 | F1, F2, F4, F7 | 9 |
| Noweir[55] | Egypt | Prospective | Byssinosis | 10 | 95/625 | F1, F4 | 5 |
| Wang et al[35] | China | Retrospective and Prospective | Silicosis | 43 | 9377/39808 | F2, F6 | 9 |
| Hnizdoand Sluis-Cremer[48] | South African | Prospective | Silicosis | 23 | 313/2235 | F6 | 7 |
| Guan et al[18] | China | Retrospective and Prospective | Silicosis | 44 | 316/3147 | F1, F2, F5, F7 | 9 |
| Zhang et al[56] | China | Retrospective | Silicosis | 35 | 1769/9268 | F5 | 6 |
| Zhang and Wang[42] | China | Retrospective | CWP | | 3224/21000 | F4 | 7 |
| Liu[57] | China | Retrospective | CWP | 7 | 1624/15647 | F3, F4 | 5 |
| Poinen-Rughooputh et al[8] | China | Prospective | Silicosis | 54 | 1219/8887 | F1 | 7 |
| Rosenman et al[58] | U.S. | Prospective | Silicosis | 5.5 | 92/937 | F2, F4, F6 | 7 |
| Han et al[10] | China | Retrospective | CWP | 48 | 411/18705 | F3 | 8 |
| Liu et al[40] | China | Retrospective | CWP | 50 | 236/16154 | F3, F4 | 8 |
| Cui et al[6] | China | Retrospective and Prospective | CWP | 44 | 2873/87904 | F3 | 7 |
| Shen et al[41] | China | Retrospective and Prospective | CWP | 41 | 1847/45589 | F3, F5 | 8 |
| Shen et al[47] | China | Retrospective | CWP | 42 | 838/17023 | F3, F4, F5 | 9 |
| Steenland and Brown[49] | USA | Retrospective | Silicosis | 25 | 170/3330 | F6 | 7 |
| Chen et al[59] | China | Retrospective | Silicosis | 35 | 1015/3010 | F5 | 6 |

Risk factors: F1: gender; F2: smoking; F3: coal workers' occupational categories; F4: duration of dust exposure; F5: cumulative total dust exposure; F6: cumulative silica exposure; F7: age.
CWP, Coal Worker's Pneumoconiosis; NOS, Newcastle-Ottawa Scale.

a random-effects model. No data pooling was possible for any identified subgroup (figure 3B).

## Age

Two studies reported the association between age and morbidity of pneumoconiosis. These studies[17 18] roughly suggest that the highest incidence occurs between the ages of 50 and 60. When the age is below 50 years old, the incidence rate increases with age, whereas when the age is above 60 years old, the incidence rate decreases with age.

## Occupational category

Since occupational categories were only reported in CWP-related studies, we summarised the relationship between occupational categories and the incidence rate of pneumoconiosis in these studies. In six studies involving 220911 participants and 8791 patients, a comparison was conducted between four distinct occupational categories. The incidence was highest among

tunnelling workers (0.112, 95% CI 0.062 to 0.174) and lowest among helping workers (0.005, 95% CI 0.001 to 0.019), according to the pooled results from various occupational categories. Thus, we compared tunnelling workers to other categories (mining, combining and helping workers) and helping workers to workers in other categories (tunnelling, mining and combining workers). Using a random-effects model, the pooled results of six studies indicated that tunnelling workers had a significantly higher risk than other workers (RR 4.75; 95% CI 1.96 to 11.33; p<0.001) with heterogeneity (p<0.001, $I^2$=100%) (figure 3C). And helping workers had an obviously lower risk than other workers (RR 0.07; 95% CI 0.03 to 0.16; p<0.001) with heterogeneity (p<0.001, $I^2$=99%) (figure 3D). Significant heterogeneity and the limited number of included studies may compromise the validity of our findings.

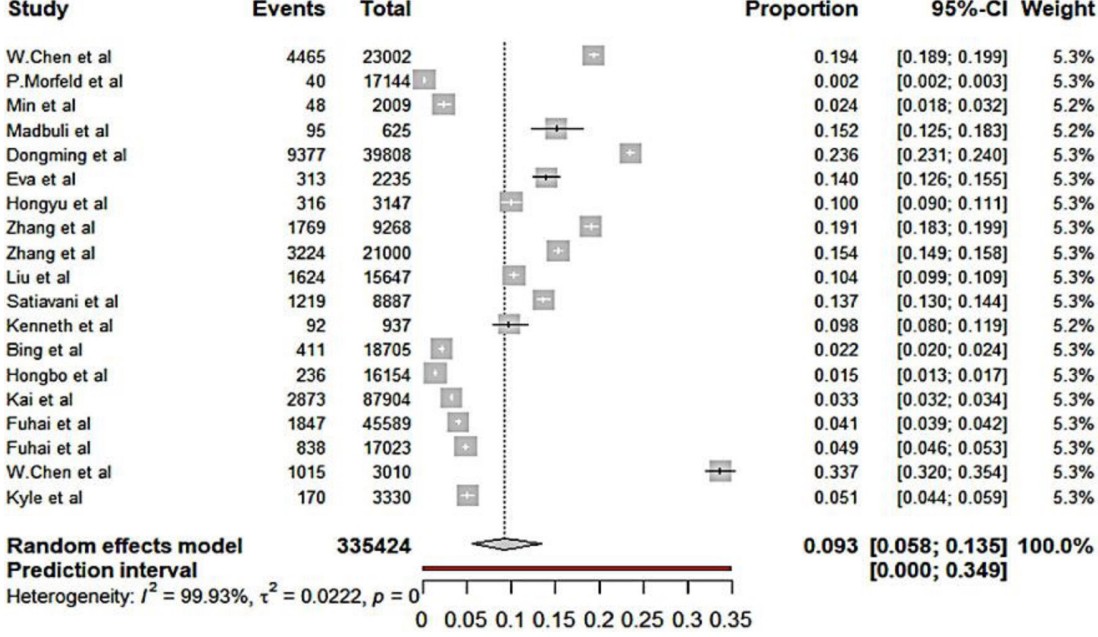

**Figure 2** The forest plot of pooled incidence of pneumoconiosis.

## Duration of dust exposure

Seven studies, including six high-quality studies and one low-quality study, reported the relationship between the duration of dust exposure and the incidence of pneumoconiosis. The summary effect size for pneumoconiosis morbidity comparing the longest and shortest duration of dust exposure was calculated by using a random-effects model (RR 4.59, 95% CI 2.41 to 8.74, p<0.01; online supplemental figure 3), indicating a significant association; however, evidence of high heterogeneity was found between studies ($I^2$=91.49%, p<0.01). In dose–response analysis, there was no indication of a nonlinear association between the duration of dust exposure and risk of pneumoconiosis (P-non-linearity=0.10; figure 4A).

Furthermore, linear dose-response meta-analysis showed that an increase in duration by 5 years was positively associated with a 26% higher risk of pneumoconiosis (RR 1.26, 95% CI 1.15 to 1.39, p<0.01; figure 4C) with heterogeneity ($I^2$=97.68%, p<0.01), regarding 2.5 years as a reference. No data pooling was possible for dose-response meta-analysis of any identified subgroup.

## Cumulative total dust concentration

Six studies involving a total of 189 037 participants and 10 271 patients reported the association between CTD and pneumoconiosis incidence. The summary effect size for pneumoconiosis morbidity comparing the highest and lowest CTD was calculated by using a random-effects

**Table 2** The pooled results and subgroup analysis of the incidence of pneumoconiosis

| | | No of studies | n/N | Pooled incidence rate | | | Heterogeneity | |
| | | | | Effects size | Lower limit | Upper limit | $I^2$ | P value |
|---|---|---|---|---|---|---|---|---|
| Total | | 19 | 29 972/335 424 | 0.093 | 0.058 | 0.135 | 99.93% | <0.001 |
| Pneumoconiosis | Silicosis | 11 | 18 824/112 777 | 0.119 | 0.064 | 0.188 | 99.92% | <0.001 |
| | CWP | 7 | 11 503/222 022 | 0.044 | 0.024 | 0.082 | 99.90% | <0.001 |
| Gender | Male | 5 | 3112/19 946 | 0.034 | 0.000 | 0.123 | 98.96% | <0.010 |
| | Female | 5 | 972/12 462 | 0.012 | 0.002 | 0.086 | 98.45% | <0.010 |
| Smoker | Yes | 5 | 4845/35 677 | 0.064 | 0.012 | 0.152 | 99.93% | <0.001 |
| | No | 5 | 1388/16 479 | 0.019 | 0.005 | 0.076 | 98.88% | <0.010 |
| Occupational categories of coal workers | Tunnelling | 6 | 3675/36 573 | 0.112 | 0.062 | 0.174 | 99.63% | <0.010 |
| | Mining | 6 | 1508/21 948 | 0.059 | 0.025 | 0.093 | 99.06% | <0.010 |
| | Combining | 6 | 2861/42 886 | 0.049 | 0.020 | 0.123 | 99.82% | <0.001 |
| | Helping | 6 | 765/119 854 | 0.005 | 0.001 | 0.019 | 99.56% | <0.010 |

CWP, coal worker's pneumoconiosis.

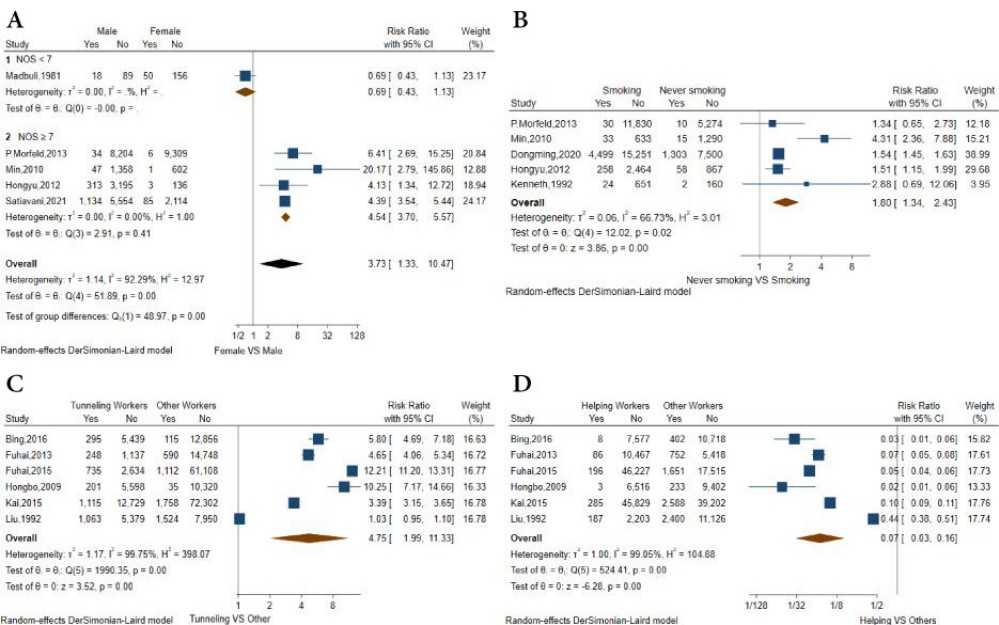

**Figure 3** Forest plot detailing risk ratios (RR) for the influencing factors when comparing participants who developed pneumoconiosis with controls. 'Yes' indicates cases who developed pneumoconiosis; 'no' indicates cases who did not develop pneumoconiosis. (A) Male versus female (RR 3.73, 95% CI 1.33 to 10.47) with the subgroup analysis of study quality (NOS <7 vs NOS ≥7), (B) smoking versus nerve smoking (RR 1.80, 95% CI 1.34 to 2.43), (C) tunnelling versus others (RR 4.75, 95% CI 1.99 to 13.33), (D) helping versus others (RR 0.07, 95% CI 0.03=0.16). NOS, Newcastle Ottawa Scale.

model (RR 34.14, 95% CI 17.50 to 66.63, p<0.01; online supplemental figure 4), indicating a significant association; however, there was evidence of high heterogeneity was found among studies ($I^2$=95.70%, p<0.01). After pooling six study-specific estimates using the restricted maximum likelihood method in a random-effects model, the result of the formal statistical hypothesis test indicated that there was no indication of a nonlinear association between the duration of dust exposure and pneumoconiosis risk (P-non-linearity=0.16; figure 4b). Furthermore, linear dose-response meta-analysis showed that a 50 mg/m³ years increase in CTD was associated with a 38% higher risk of pneumoconiosis (RR 1.38, 95% CI 1.22 to 1.55, p<0.01; figure 4D) with heterogeneity ($I^2$=99.75%, p<0.01), regarding 50 mg/m³ years as a reference.

Different sorts of risk values retrieved from each study were determined to be included in the analysis during the data analysis. Only one of the six studies had an adjusted RR. Therefore, this study was eliminated for sensitivity analysis. Sensitivity analysis for the link between CTD and the risk of pneumoconiosis revealed that excluding the study by Guan HY *et al*[18] did not significantly modify the estimates, either in the shape of association (P-non-linearity=0.74; online supplemental figure 5) or the pooled effect sizes (detailed in online supplemental figure 6 and online supplemental figure 7).

### Cumulative silica exposure
Four studies reported the association between CSE and the incidence of silicosis, which involved 40 212 participants and 6371 patients. The Pearson correlation analysis revealed that the incidence of silicosis and CSE were highly correlated (r=0.794, p<0.001, figure 5). No data pooling was possible for regression analysis of any identified subgroup.

### DISCUSSION
This is, to our knowledge, the first systematic review and meta-analysis to estimate the incidence of pneumoconiosis in dust-exposed workers and the relationship between pneumoconiosis and various influencing factors. Our meta-analysis revealed that the incidence among workers exposed to dust was approximately 9.3% (95% CI 8.5% to 13.5%). And the effect sizes of influencing factors, including gender, smoking, age, occupational categories, duration of dust exposure, CTD and CSE, was investigated using a larger sample size and a more comprehensive methodology than ever before. Further, in terms of the quantified relationship between risk factors and the incidence, we found a linear dose–response relationship between CTD and exposure duration and the pneumoconiosis risk, and a linear correlation between CSE and the incidence of silicosis.

The pooled incidence estimated by this meta-analysis was higher than the Global Burden of Disease Study 2017 analysis's reported incidence rate of 7.5% (95% CI 6.6% to 8.4%).[19] The first possible reason for it was that the majority of studies included in our meta-analysis originated from nations with relatively higher incidence rates of pneumoconiosis, such as China and America. Second, the incidence of pneumoconiosis has been declining recently as preventive measures have improved. It is also possible that the pooled incidence is higher than

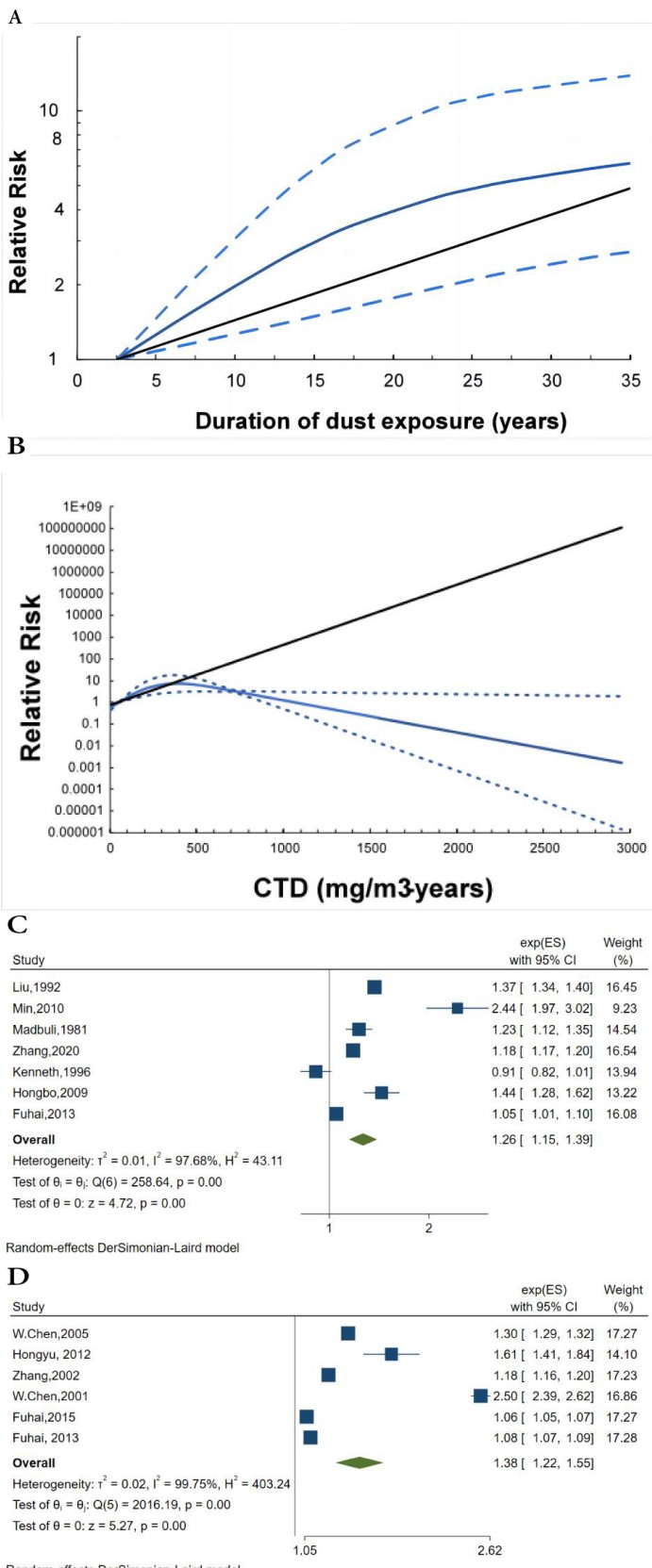

**Figure 4** The association between the risk of pneumoconiosis and dust exposure. The black line indicates the linear model; the sky-blue line indicates the spline model; the dashed lines represent 95% CIs. (A) The dose–response association between the duration of dust exposure and the risk of pneumoconiosis (P-non-linearity: 0.10), (B) The dose–response association between the CTD and the risk of pneumoconiosis (P-non-linearity: 0.16), (C) Forest plot detailing risk ratios (RR) for each 5-year increase in the duration of dust exposure (RR 1.26, 95% CI 1.15 to 1.39), (D) Forest plot detailing RRs for each 50 mg/m³-years increase in CTD (RR 1.38, 95% CI 1.38 to 1.55). CTD, cumulative dust concentration; ES, effect size.

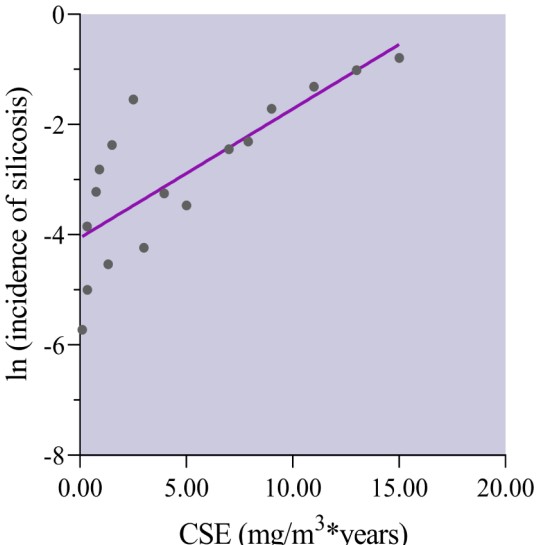

**Figure 5** Correlation between cumulative silica exposure (CSE) and incidence of silicosis. Concentrations of CSE and the incidence were highly correlated (r=0.794, p<0.001). Y scales are logarithmically transformed.

the incidence or prevalence reported by recent studies because the type of article included in this meta-analysis is a cohort study. This is based on the observation that our pooled CWP incidence differs significantly from that reported by the analysis from the Global Burden of Disease Study 2017 (1.9%, 95% CI 1.5% to 2.5%),[19] but is comparable to that (3.7%, 95% CI 3.0% to 4.5%) of a meta-analysis for CWP among underground miners.[20] In comparison to recent studies, the pooled results of only cohort studies typically allowed for larger sample size and greater access to the true conclusion.

In addition, our results demonstrated that the pooled incidence of silicosis is higher than that of CWP, which is consistent with the findings of numerous previous studies. Numerous pathological processes, including macrophage autophagy, autoimmune, apoptosis and proptosis, have indicated that crystalline silica or quartz, the most prevalent occupational dust, is the most significant cause of silicosis.[21–23] However, rather than silica or quartz, the amount of carbon in the coal is what largely determines the risk of CWP.[24–26] In other words, the incidence of silicosis is higher than that of CWP, possibly because the pathogenic exposures that cause silicosis are more common than those that cause CWP. Nevertheless, the mechanism that perfectly explains the epidemiological data remains elusive.

Consistent with many previous studies,[8 17 27] our findings showed a difference in the pooled incidence of pneumoconiosis between men and women, with men being at higher risk than women. On the other hand, some previous studies have shown no difference in the incidence of pneumoconiosis between men and women,[28–30] however, limitations such as cross-sectional studies, medical examination and a short period of follow-up rendered their findings inconclusive. There are several

possible explanations for the difference in pneumoconiosis risk between men and women. First, the greater physical strength of men during labour contributes to more rapid pulmonary ventilation, which in turn results in the inhalation of a significantly higher concentration of occupational dust over the same period when compared with women.[31 32] Second, as demonstrated by previous gender-related research, some gender disparities are largely mediated by differences in the expression of genotypes, such as differential expression of proinflammatory cytokines.[33 34] Third, a higher proportion of male smokers compared with female smokers was associated with an increased risk of pneumoconiosis in the male group.

Concerning the relationship between smoking and pneumoconiosis, the current studies primarily focus on the morbidity and mortality of pneumoconiosis,[35 36] and it is generally accepted that smoking can promote the occurrence of pneumoconiosis and also increase all-cause mortality in patients with pneumoconiosis, despite the fact that the risk for smoking varies widely between studies. In addition, Wang et al[35] found that as the dust concentration rises, the incidence of pneumoconiosis increases more rapidly in smokers than in nonsmokers. Previous animal research demonstrated that exposure to tobacco smoke alone had a minimal effect on these toxicity parameters, however, the combined exposure to tobacco smoke and occupational dust worsened the lung response in comparison to exposure to either agent alone.[37] One possible explanation is that cigarette smoking destroys the ciliated epithelium in the bronchi and reduces the clearance of dust from the lungs. Moreover, Dorman et al[38] discovered that exposure to tobacco smoke exacerbated the silica dust-induced lung toxicity in rats by altering the level of secretion of several cytokines/chemokines and oxidants produced by the bronchoalveolar phagocytes. Alternately, exposure to a toxic agent may result in changes in gene expression accompanied by alterations in the functions mediated by the affected genes, resulting in toxicity to the target organ. SPP1, a gene involved in lung inflammatory and fibrotic responses to toxic particles, was overexpressed by 2.85-fold, 36.69-fold and 72.61-fold, in the smoke, silica and smoke plus silica exposed rats, respectively, compared with the air controls.[39] Currently, it is imperative that smokers quit, and they may require lower dust concentrations and stricter prevention and control of dust inhalation than nonsmokers. However, the effects of tobacco smoke and occupational dust on the lungs are complex, and further study is required to determine how they interact.

There are occupational categories for any worker who is exposed to dust on the job, but the vast majority of occupational studies on this topic have focused on coal miners.[6 40 41] The average dust concentration in the workplace is one of the possible primary causes of disparate morbidity. Cui et al[6] found that the incidence of pneumoconiosis in the same occupational category in various coal mines was proportional to the average dust concentration; that is, the incidence increased as the average dust

concentration increased. There are, of course, numerous other distinctions between occupational categories, such as working hours, work habits and lifestyles. Whether these factors are also the reasons for the different incidence rates between them needs to be discovered and confirmed by future research.

The observed positive association is consistent with results from previous clinical studies on the duration and incidence of pneumoconiosis. In 11 state-owned coal mines, for instance, a retrospective cohort study found that the rate of morbidity increased sharply with the length of employment.[42] Since the majority of studies on the risk of different lengths of employment have been measured in increments of 5 years, our study also calculated the risk increase for each additional 5 years of employment. Some reviews[43–46] proposed that the development and progression of pneumoconiosis are chronic inflammatories, anti-inflammatory and fibrotic processes, indicating that pneumoconiosis progresses gradually under the continuous stimulation of occupational dust. So, we believe that reducing the duration of employment in an environment with a high dust concentration or implementing targeted shift changes for workers with an overall increase in morbidity. However, the efficacy and mechanism of these targeted preventive measures must be confirmed through additional research.

Although there is no comparable meta-analysis for CTD, our results are consistent with previous clinical research that revealed a positive association between the incidence of pneumoconiosis and CTD.[6 9 18 47] Present studies suggested that dust is still the primary occupational hazard for developing pneumoconiosis; therefore, if dust concentrations increase, pneumoconiosis risk should theoretically increase as well.[47] We believe that CTD is a more significant method for evaluating exposure than either the average dust concentration or the duration of exposure because it takes both into account. However, we detected further risk growth beyond the increment of 50 mg/m$^3$ years in this meta-analysis, due to a reduction in sample size, the relationship between a higher concentration of CTD and the risk of pneumoconiosis is less reliable. The association between CTD and the incidence of pneumoconiosis should be further investigated and analysed, and the Occupational Safety and Health Administration should eventually impose permissible limits for CTD.

In the absence of the effect size of silica on the risk of silicosis, we estimated the association between CSE and the incidence of silicosis using Pearson correlation analysis. Previous studies have identified a positive correlation between the incidence of silicosis and silica.[35 48 49] For example, Souza TP *et al* found that the incidence ratio of silicosis increased by 4% for each additional year of silica exposure.[50] However, CSE is a more accurate method of assessing silica exposure than duration because it takes dust concentration into account. There is no previous study to explore the linear association between CSE and the incidence of silicosis because of fewer exposures'

stratification in each study. In our meta-analysis, we found a positively strong linear correlation. At present, most countries regulate exposure limits for respirable crystalline silica in the range of 0.05–0.1 mg/m$^3$.[51] However, this occupational exposure standard is too strict for most workplaces, and they may exceed it for brief periods.[52] Moreover, new information suggests that it would be more effective to control the risk of pneumoconiosis by imposing a more accurate assessment of dust exposure, as measured in hours, days, weeks, or months.[53] We believed that our findings could provide some insights into this new understanding.

## Limitations

There are some inevitable limitations in our meta-analysis. First, significant heterogeneity in the pooled results for a variety of reasons and the small number of included studies may reduce the validity of our results for certain risk factors. Second, given the limited literature available, some potentially relevant risk factors, such as alcohol consumption and tuberculosis, were not included in this meta-analysis.

## CONCLUSIONS

In conclusion, this meta-analysis revealed that the incidence of pneumoconiosis among occupational workers was 0.093 and that there were seven factors associated with this incidence, including male, smoking, age, coal workers' occupational categories, duration of dust exposure, CTD and CSE. In addition, we discovered a linear dose–response relationship between the risk of pneumoconiosis and the duration of dust exposure and CTD. Our findings will provide hints for pneumoconiosis prevention strategies.

**Acknowledgements** The authors would also like to thank Professors from the College of Public Health, Shanxi Medical University for the advice on the study design and revise opinion.

**Contributors** XS, XY and XK completed the search, determined eligible papers for inclusion, and completed the quality assessment, and data extraction. XS and XZ completed the meta-analysis. All authors contributed to the writing of the final manuscript. XZ is the guarantor of the paper.

**Funding** Organisation: (1)The National Health Commission Key Laboratory of Pneumoconiosis (Shanxi, China) Project; Project number: No. 2020-PT320-005; Project Leader: XZ. (2) Key Laboratory of Respiratory Disease Prevention and Control of Shanxi.

**Competing interests** None declared.

**Patient and public involvement** Patients and/or the public were not involved in the design, or conduct, or reporting, or dissemination plans of this research.

**Patient consent for publication** Not applicable.

**Provenance and peer review** Not commissioned; externally peer reviewed.

**Data availability statement** All data relevant to the study are available in the article, online supplemental information, or a public, open access repository. Characteristics of included studies, details of literature quality assessment, and raw data used for analysis have been submitted to and made available by the Dryad repository on request. Extra data can be accessed via the Dryad data repository at http://datadryad.org/ with the doi: 10.5061/dryad.rv15dv4c3.

peer-reviewed. Any opinions or recommendations discussed are solely those of the author(s) and are not endorsed by BMJ. BMJ disclaims all liability and responsibility arising from any reliance placed on the content. Where the content includes any translated material, BMJ does not warrant the accuracy and reliability of the translations (including but not limited to local regulations, clinical guidelines, terminology, drug names and drug dosages), and is not responsible for any error and/or omissions arising from translation and adaptation or otherwise.

**ORCID iD**
Xuesen Su http://orcid.org/0000-0003-4905-6200

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
