## [Reviewer comments · BMJ Open]

ARTICLE DETAILS

TITLE (PROVISIONAL)	Incidence and influencing factors of occupational pneumoconiosis: A systematic review and meta-analysis.
AUTHORS	Su, Xuesen; Kong, Xiaomei; Yu, xiao; Zhang, Xinri

VERSION 1 – REVIEW

REVIEWER	Zhao, Kun Zhejiang University
REVIEW RETURNED	23-Jun-2022

GENERAL COMMENTS	This manuscript performed a meta-analysis to estimate the prevalence and incidence and influencing factors of pneumoconiosis. This is a meaningful topic and will contribute to the prevention of pneumoconiosis. But there is some suggestions and hope can improve it. Abstract: 1. In Participants section, in general we do not show results in the methodology section, please remove “Finally, 19 studies met the inclusion and exclusion criteria and were included in our meta-analysis, involving 335,424 participants, of whom 29,972 developed pneumoconiosis.” to result section.2. The incidence of silicosis was highly collinear with CSE ($r=0.794$, $P<0.001$), and an additional 1 mg/m³ -years of CSE causes an increase in incidence of silicosis by 2.19% (95%CI: 1.00%~4.86%). Why did not mention it in abstract but in Article Summary? Method: 3. Page 7, Line13-14, what is the reference of “the national diagnostic criteria of pneumoconiosis”.4. Page 8, Line 15-16, risk estimates with 95% CIs from any statistic model. How to deal with the original data, adjustment data or after propensity match?5. Page 9, line 18, A two stage or at second stage? Results: 6. Why choose these seven factors? Please define the criteria.7. The number of figures is too many and whether they meet the requirement of journal? Discussion 8. Please summary the evidence in the first paragraph, it should perform according to PRISMA statement.9. Why write the paragraph, This meta-analysis summarized several influencing factors with a larger sample size and a more comprehensive way ever before, and explored effect size of influencing factors on developing pneumoconiosis, including male, smoking, age and occupational categories.10. In the section of discussion, please don't repeat the result of estimate.
--

	Minor comments; Some Format error needs to be corrected, such as space is required after comma (page 11, line 19; page 17 line 7)
--	--

REVIEWER	Xie, Yang The First Affiliated Hospital of Henan University of Chinese Medicine, Department of Respiratory Diseases
REVIEW RETURNED	27-Jun-2022

GENERAL COMMENTS	I have examined this article carefully with great interest, however, some shortcomings should be revised :  1、 “Objectives: To estimate the prevalence and incidence of pneumoconiosis worldwide and explore influencing factors of it.” This study did not explore the incidence of pneumoconiosis. 2、 “19 studies met the inclusion and exclusion criteria and were included in our meta-analysis”, 19 studies met the exclusion criteria, Is it incorrect? 3、 “searched the literature published before November 2021“ and “with no limitation on time of publication” , The two sentences were repeated. 4、 In “Data extraction”, “duration of follow-up” was repeated. 5、 In “Influencing factors of pneumoconiosis”, “Some evidence of heterogeneity was detected between subgroups in stratified analyses for high quality studies (RR =4.54, 95% CI 3.70~5.57, P<0.00001) with heterogeneity (P=0.4, I2=0%) (Fig. 5).” P=0.4, I2=0%, there was no heterogeneity. 6、 In “Duration of dust exposure” and “Cumulative total dust concentration (CTD)”, Please explain the significance of RR values(RR:1.26, RR:1.38) in the linear dose-response meta-analysis. 7、 “This is based on the fact that our pooled prevalence of CWP (4.4%, 95 % CI: 2.4%~8.2%) differ greatly from those of this report about CWP (1.9%, 95%CI: (1.5%~2.5%), but are close to those (3.7%, 95% CI 3.0~4.5%) of a meta-analysis for CWP among underground miners”, “this report about CWP (1.9%, 95%CI: (1.5%~2.5%)” means which report? 8、 In “Limitations”, “considering the limited literature available, some potentially relevant risk factors were not included in this meta-analysis, such as age, alcohol drinking and tuberculosis.” Whether the age factor was included in the meta-analysis. 9、 In the results section of the abstract, the results of the cumulative silica exposure and age should also be described. 10、 There are grammatical and editorial issues in this paper. We recommend that you continue to optimize the expression of the language in order to improve the overall quality of the article
--

REVIEWER	Emir, Büşra Izmir Katip Celebi Universitesi
REVIEW RETURNED	22-Aug-2022

GENERAL COMMENTS	An overall interesting systematic review that aims to address and explore the prevalence, the impact of factors on the morbidity of pneumoconiosis. I suggest reviewing and clarifying some items.  1. Page 15 of 65, row 25 In the result section, titled “Pooled results, sensitive analysis, publication bias of the prevalence of pneumoconiosis”
---

	“Based on the results of random-effects method, the pooled prevalence of pneumoconiosis among dust exposed workers was 0.093 (95% CI: 0.085~0.135), with a high-level between-study heterogeneity ($I^2 = 99.93\%$, $p < 0.0001$) (Fig.2, Table 2).” 95% CI lower and upper limits notation of “~” Between the lower and upper limit values of the 95% confidence interval, a “-” sign should be placed instead of the “~” tilde. The “-” notation should be preferred over the tilde for all confidence interval representations in manuscript. 2. The caption of the figures 5-6 and the forest plot expression in the manuscript are used incorrectly. It should be corrected as a Forest plot in the relevant parts. 3. Fig 10 y axis “Relative Risk” not “Ralative Risk” 4. How do the authors explain dose-response association of CTD with risk of pneumoconiosis? CTD was modelled with restricted cubic splines in a multivariate random effects dose-response model. However, it was stated in Figure 14 that there was a linear relationship later. I suggest adding a more detailed paragraph about it.
--	---

REVIEWER	Chinchilli, Vernon M. The Pennsylvania State University, Public Health Sciences
REVIEW RETURNED	23-Oct-2022

GENERAL COMMENTS	The authors only used two terms in their literature search, namely, “pneumoconiosis” and “cohort studies.” It is surprising that they did not consider other terms, such as “dust inhalation,” “occupational dust,” and “silicosis.” The authors seem to use “prevalence” and “incidence” interchangeably, which they should not do. If they only identified cohort studies, then it is assumed that the cohort studies estimated incidence because they only investigated new cases of pneumoconiosis. The authors state that a cohort study design was an inclusion criterion for the meta-analysis, but they included some retrospective studies. The authors have applied appropriate statistical methodology for the meta-analysis. The authors should seek assistance with the writing and grammatical structure in the manuscript. There are 15 figures. The authors should identify some of them for a supplemental section.
--

REVIEWER	Cai, Jingheng Sun Yat Sen Univ, Statistics
REVIEW RETURNED	23-Oct-2022

GENERAL COMMENTS	The manuscript is aimed to estimate the prevalence and incidence of pneumoconiosis worldwide and explore the related risk factors. Given that I am not an expert of medical field, I can only comment on the statistical analysis part. The authors employed several models to analyze the data, however, some models and results were not presented clearly.
--

	1. Page 9, last paragraph. It seems that the authors employed restricted cubic spline models to assess the general relationships between pneumoconiosis and its related risk factors, and the results were given in Pages 17 and 18. As the models are complicated and not presented clearly, it is difficult to understand the results. I suggest the authors give the details of the model (maybe in the appendix). 2. Page 10, second paragraph. The authors used a linear regression to examine the effects of CSE on incidence of silicosis. However, since the response variable is the incidence rate of silicosis, ordinary linear regression is not suitable even with the logarithm transformation. Generalized linear models are more appropriate. 3. The writing is poor, necessary edit is needed.
--	--

VERSION 1 – AUTHOR RESPONSE

Responds to the reviewer’s comments:

Reviewer #1: Dr. Kun Zhao

1. Response to comment: please remove “Finally, 19 studies met the inclusion and exclusion criteria and were included in our meta-analysis, involving 335,424 participants, of whom 29,972 developed pneumoconiosis.” to the result section.

Response: We moved the part suggested by the reviewer to the abstract's results section on **page 2, lines 14-15**.

2. Response to comment: Why did not mention the results of CSE in the abstract but in Article Summary?

Response: Because we considered including this section in the results, the abstract's length may have increased. However, the outcomes of CSE are also significant. So, we completely agreed with the reviewer and included this section in the abstract with a minor length increase, addressed on **page 3, lines 3-4**.

3. Response to comment: What is the reference to “the national diagnostic criteria of pneumoconiosis”?

Response: We wholeheartedly concur with the reviewer. In response to the reviewer's question, we've included three references on international pneumoconiosis diagnostic standards here (*references 12-14*). And, on **page 7, lines 12-16**, we have further explained the diagnostic criteria, "*the national diagnostic criteria of pneumoconiosis published by the International Labour Organization (ILO) (1980, 2000, and 2009 editions) or Ministry of Health of the People's Republic of China (GB 5906-1986, GB 5906-1997, GBZ 70-2002, GBZ 70-2009, and GBZ 70-2015)*."

It should be emphasized that pneumoconiosis is a disease that has gained increasing public attention as industrial production has increased. As a result, recognition of this disease has taken a long time, resulting in numerous revisions and updates to pneumoconiosis diagnostic criteria. Because of the vast number of studies included in the meta-analysis, separate studies may employ various versions of diagnostic criteria, or the same study may use distinct versions of diagnostic criteria. However, the main material of these multiple versions of the pneumoconiosis diagnostic criteria has not changed significantly, and the different versions and revisions are all to improve the details.

4. Response to comment: How to deal with the original data, adjustment data, or after propensity match about risk estimates with 95% CIs from any statistic model?

Response: We planned to collect all of the risk estimates with 95% CIs from any statistic model in the article during the data extraction phase, group them according to whether they had been changed, and evaluate them independently. During the data analysis step, however, we discovered that five of

the six studies reported an unadjusted risk ratio and one (*reference 22*) presented an adjusted risk ratio. As a result, we removed this study for sensitivity analysis, which is addressed on **page 19, lines 9-15**, and described in *Supplementary Figs. 5, 6, and 7*.

5. Response to comment: Page 9, line 18, A two stage or at second stage?

Response: Thank you for the thorough assessment by the reviewer and for identifying our bug here. In fact, in dose-response analysis, we employed two-stage random-effects linear and non-linear models, instead of “two stage”. Therefore, the following changes have been made, “*Using a two-stage random-effects dose-response meta-analysis, a potential non-linear association between the risk of pneumoconiosis and two aspects of dust exposure was investigated*”, addressed on **page 9, line 22**.

6. Response to comment: Why choose these seven factors? Please define the criteria.

Response: The criteria for selecting influencing factors are further detailed in the manuscript as follows: “*we chose influencing factors that were mentioned in at least two studies to investigate the relationship between those and the incidence of pneumoconiosis*”, addressed on **page 9, line 3**.

7. Response to comment: The number of figures is too many.

Response: The number of figures stated by the reviewer is critical, and we have decreased the number of figures to 5 by merging several figures and adding supplemental figures.

8. Response to comment: Please summarize the evidence in the first paragraph.

Response: We have summarized the evidence in the first paragraph of the discussion, addressed on **page 20, lines 1-11**.

9. Response to comment: Why write the paragraph in the discussion section, “This meta-analysis summarized several influencing factors ...”

Response: That paragraph was written to summarize the results on influencing factors, and we agreed with the comments on item 8 and made changes to this paragraph, which has been deleted.

10. Response to comment: In the section of the discussion, please don't repeat the result of the estimate.

Response: We have simplified the discussion part and omitted the outcomes of these repeated narratives.

11. Response to comment: some format error needs to be corrected.

Response: Thank you for the reminder; we double-checked the formatting of the manuscript and repaired any errors.

Reviewer #2: Dr. Yang Xie

1. Response to comment: This study did not explore the incidence of pneumoconiosis.

Response: We combined prevalence and incidence in our manuscript, which was incorrect; thank you for your suggestions. Dr. Vernon M. Chinchilli, reviewer #4, also highlighted the following about this issue: *The authors seem to use “prevalence” and “incidence” interchangeably, which they should not do. If they only identified cohort studies, then it is assumed that the cohort studies estimated incidence because they only investigated new cases of pneumoconiosis.* These two comments may appear to be contradictory.

We finally felt that the viewpoint of Dr. Vernon M. Chinchilli could be more appropriate after thorough consideration of the two reviewers' opinions and the search for relevant evidence, thus we substituted the prevalence with the incidence, including the title, abstract, and main document file. This section of the comments may have been misunderstood by us. If the reviewer has differing viewpoints, we greatly hope you can provide us with additional explanation. Thank you for your comments once more.

2. Response to comment: 19 studies met the exclusion criteria, Is it incorrect?

Response: We completely agreed with the reviewer that this expression is incorrect. We've also made the following changes: “Finally, 19 studies were included in our meta-analysis,” **page 11, lines 19-20**.

3. Response to comment: “searched the literature published before November 2021” and “with no limitation on time of publication”, The two sentences were repeated.

Response: Thank you for bringing this to our attention; we have made the following changes: “*two reviewers independently and systematically searched through online databases, including PubMed,*

EMBASE, the Cochrane Library, and Web of Science before November 2021, without regard to the language of publication.", addressed on **page 6, line 20-page 7, line 1.**

4. Response to comment: "duration of follow-up" was repeated

Response: We knew such an error could not happen. Thank you for reminding us; we have eliminated the duplicates, which were addressed on **page 8, line 12.**

5. Response to comment: In "Influencing factors of pneumoconiosis", "Some evidence of heterogeneity was detected between subgroups in stratified analyses for high quality studies (RR =4.54, 95% CI 3.70~5.57, P<0.00001) with heterogeneity (P=0.4, I²=0%) (Fig. 5)." P=0.4, I²=0%, there was no heterogeneity.

Response: The reviewer's careful review has made our article more rigorous, and we have made the following changes: "*Subgroup analyses for only high-quality studies (RR=4.54, 95%CI 3.70-5.57, P<0.00001) with non-significant heterogeneity (P=0.40, I²=0%) revealed the possible cause of significant heterogeneity (Fig. 3).*", addressed on **page 16, line 16.**

6. Response to comment: Please explain the significance of RR values (RR:1.26, RR:1.38) in the linear dose-response meta-analysis.

Response: Following the reviewer's comments, we have made the following changes:

"*an increase in duration by 5 years was associated with a 26% higher risk of pneumoconiosis (RR=1.26, 95%CI: 1.15-1.39, P<0.01; Fig. 4c) with heterogeneity (I²=97.68%, P<0.01), regarding 2.5 years as a reference.*", addressed on **page 18, lines 12-15.**

"*a 50 mg/m³-years increase in CTD was associated with a 38% higher risk of pneumoconiosis (RR=1.38, 95%CI: 1.22-1.55, P<0.01; Fig. 4d) with heterogeneity (I²=99.75%, P<0.01), regarding 50 mg/m³-years as a reference.*", addressed on **page 19, lines 5-8.**

7. Response to comment: "This is based on the fact that our pooled prevalence of CWP (4.4%, 95 % CI: 2.4%~8.2%) differ greatly from those of this report about CWP (1.9%, 95%CI: (1.5%~2.5%), but are close to those (3.7%, 95% CI 3.0~4.5%) of a meta-analysis for CWP among underground miners", "this report about CWP (1.9%, 95%CI: (1.5%~2.5%)" means which report?

Response: This report refers to the 2017 Global Burden of Disease Study analysis. And to express the original intention more accurately, we have made the following changes: "*This is based on the observation that our pooled CWP incidence differs significantly from that reported by an analysis from the Global Burden of Disease Study 2017 (1.9%, 95%CI: 1.5%-2.5%), but is comparable to that (3.7%, 95%CI: 3.0%-4.5%) of a meta-analysis for CWP among underground miners*", addressed on **page 20, lines 20-22.**

8. Response to comment: In "Limitations", whether the age factor was included in the meta-analysis.

Response: We completely agreed with the reviewer and have made the following modifications: "*Second, given the limited literature available, some potentially relevant risk factors, such as alcohol consumption and tuberculosis, were not included in this meta-analysis.*" (**Page 25, lines 17-18).**

9. Response to comment: the results of the cumulative silica exposure and age should also be described in the abstract.

Response: We entirely agreed with and appreciated the reviewer's suggestions, and we have described the age and CSE-related results in the abstract, which can be found on **pages 2 and 3 (lines 19-20 and 3-4, respectively).**

10. Response to comment: There are grammatical and editorial issues in this paper.

Response: We modified the full text with sentence processing, grammar correction, and mistake formatting.

Reviewer #3: Dr. Büşra Emir

1. Response to comment: the "-" notation should be preferred over the tilde for all confidence interval representations in the manuscript.

Response: Thank you for the comments. Following the reviewer's recommendation, we have replaced the "~" tilde with the "-" sign between the lower and upper limit values of the 95% confidence interval of the full text.

2. Response to comment: It should be corrected as a Forest plot in the relevant parts in the caption of figures 5-6

Response: Following the comment, we revised the figure captions and rechecked them to prevent this low-level problem from occurring again.

3. Response to comment: Fig 10 y axis "Relative Risk" not "Ralative Risk"

Response: Following the reviewer's reminder, we changed and rechecked the figure captions to prevent this low-level problem from occurring again. It is worth noting that the original *Fig. 10* has been replaced by *Fig. 4a* in the revised version of the manuscript.

4. Response to comment: How do the authors explain the dose-response association of CTD with the risk of pneumoconiosis? CTD was modeled with restricted cubic splines in a multivariate random effects dose-response model. However, it was stated in Figure 14 that there was a linear relationship later. I suggest adding a more detailed paragraph about dose-response analysis.

Response: To achieve the goal of effective prevention or intervention of the outcome, it is frequently important in epidemiological research to understand the potential relationship between the change of a specific amount of exposure (intervention) and the risk of outcome indicators. This is known as the dose-response relationship.

We employed two-stage random-effects non-linear models to conduct a dose-response analysis in this meta-analysis. First, we used a generalized least squares trend estimation method to produce study-specific estimates under a restricted cubic spline model. Second, we pooled study-specific results in a multivariate random-effects model using the restricted maximum likelihood method. Then, we calculated a probability value for non-linearity (P-nonlinearity) using null hypothesis testing, with the regression coefficient of the second spline set to zero (indicating a linear dose-response analysis). P-nonlinearity was 0.16, showing that the null hypothesis, a linear dose-response association between CTD and the risk of pneumoconiosis, was accepted.

We sought to further quantify this link when the results revealed a linear dose-response relationship between CTD and the risk of pneumoconiosis. The initial outcome of the dose-response analysis, however, was a change in the risk of pneumoconiosis for every 1 mg/m³-year change in CTD, and a change of 1 mg/m³-years was too tiny for CTD. For the reasons stated in the discussion section, we selected to calculate the change in risk of pneumoconiosis for every 50 mg/m³-year rise in CTD. According to our study and calculations, the risk of pneumoconiosis increases by 1.38 (95% CI: 1.22-1.55) with every 50 mg/m³-year increase in CTD. The forest plot in Fig. 4d depicted the change in pneumoconiosis risk for every 50 mg/m³-year rise in CTD in each trial, as well as the overall results using this linear dose-response model. We amended the Method section on dose-response analysis (**page 9, line 22-page 10, line 16**) and provided *Supplementary Materials (The detailed description of the dose-response analysis process)* linked to the thorough description of it to clearly describe the procedure.

To more clearly explain the significance of the dose-response association between CTD and the risk of pneumoconiosis, we have made the following changes:

"a 50 mg/m³-years increase in CTD was associated with a 38% higher risk of pneumoconiosis (RR=1.38, 95%CI: 1.22-1.55, P<0.01; Fig. 4d) with heterogeneity (I²=99.75%, P<0.01), regarding 50 mg/m³-years as a reference", addressed on **page 19, lines 5-8**.

As well as the dose-response association between the durations and risk of pneumoconiosis, *"an increase in duration by 5 years was associated with a 26% higher risk of pneumoconiosis (RR=1.26, 95%CI: 1.15-1.39, P<0.01; Fig. 4c) with heterogeneity (I²=97.68%, P<0.01), regarding 2.5 years as a reference."*, addressed on **page 18, lines 12-15**.

We show both the splines and the lines in the updated *Fig. 4* to provide readers and reviewers with a better grasp of the dose-response analysis process, and the statistically significant line in the graph is black.

Reviewer #4: Dr. Vernon M. Chinchilli

1. Response to comment: The authors only used two terms in their literature search, namely, "pneumoconiosis" and "cohort studies." It is surprising that they did not consider other terms, such as "dust inhalation," "occupational dust," and "silicosis."

Response: "By using search terms related to 'pneumoconiosis' and 'cohort studies,'" says the section on Search Strategy, implying that pneumoconiosis and cohort studies are two key points or key aspects that we consider in our search strategy, rather than search words that are directly used to develop search strategies. As a result, the terms "related to" were crucial. Other terms indicated in the comments by the reviewer, such as "dust inhalation," "occupational dust," and "silicosis," are covered in the search strategy, which is listed in *Supplementary Table 1*, Search strategy of each online database.

2. Response to comment: If they only identified cohort studies, then it is assumed that the cohort studies estimated incidence because they only investigated new cases of pneumoconiosis.

Response: Thank you for pointing out that we used "prevalence" and "incidence" interchangeably in our manuscript, which was incorrect. Dr. Yang Xie, reviewer #2, also addressed this issue: "*Objectives: To estimate the prevalence and incidence of pneumoconiosis worldwide and explore influencing factors of it.*" *This study did not explore the incidence of pneumoconiosis.* These two comments may appear to be contradictory.

We finally agreed that this statement is more acceptable after thorough consideration of the two reviewers' comments and the search for relevant evidence, therefore we replaced the prevalence with the incidence, including the title, abstract, and main document file. Thank you for your comments once more.

3. Response to comment: The authors state that a cohort study design was an inclusion criterion for the meta-analysis, but they included some retrospective studies.

Response: In our meta-analysis, "retrospective studies" refers to retrospective cohort studies, which are a form of cohort research. As a result, when we restricted our search to cohort studies, we also looked for retrospective cohort studies.

4. Response to comment: The authors should seek assistance with the writing and grammatical structure of the manuscript.

Response: We completely agreed with the reviewer that good writing and proper grammatical structure are essential, so we changed our text with the help of writing and editing.

5. Response to comment: The authors should identify some of the figures for a supplemental section.

Response: The number of figures stated by the reviewer is critical, and we have decreased the number of figures to 5 by merging several figures and adding supplemental figures.

Reviewer #5: Dr. Jingheng Cai

1. Response to comment: I suggest the authors give the details of the model (maybe in the appendix)

Response: We intended to determine the shape of the association and quantify it in the section on the relationship between the risk of pneumoconiosis and CTD and the duration of dust exposure. As a result, we used dose-response analysis. Dose-response models can be either linear or nonlinear.

Introduction for restricted cubic spline models

It is difficult to choose a nonlinear model for dose-response analysis. After thoroughly reading the technique article, we discovered that the majority of regularly used models are based on parametric methods, such as restricted cubic spline models. Orsini was the first to apply restricted cubic spline to a fitted dose-response meta-analysis model (reference: *Orsini N, Li R, Wolk A, Khudyakov P, Spiegelman D. Meta-analysis for linear and nonlinear dose-response relations: examples, an evaluation of approximations, and software. Am J Epidemiol. 2012 Jan 1;175(1):66-73. doi: 10.1093/aje/kwr265. Epub 2011 Dec 1. PMID: 22135359; PMCID: PMC3244608*). This is "a more flexible restricted cubic splines method that can be used to assess nonlinearly graphically and through

a formal statistical hypothesis test," according to his explanation. The cubic spline function is the most commonly used model in dose-response meta-analysis today, and the fit is so strong that the Orsini team has published programming code for a different software, which can be found at <http://www.imm.ki.se/biostatistics/glst/>. We also cited "Two-stage random-effects linear and non-linear models using Stata" which is part of the code.

The essence of restricted cubic spline models is smooth piecewise polynomial functions. We define a function $f(x)$, which is a cubic polynomial in each subinterval of its domain and is continuous at the derivatives of order 2 and below of the subinterval, i.e., forming a cubic spline function, and the connection points between these subintervals are knots. As described in the method section, we have chosen the commonly used three knots method. Smooth piecewise polynomial functions are at the heart of restricted cubic spline models. We define a function $f(x)$ that is a cubic polynomial in each subinterval of its domain and is continuous at derivatives of order 2 and below, making a cubic spline function, and the connection points between these subintervals are nodes. We chose the popular three knots method, as mentioned in the method section.

A detailed description of the analysis process (e.g., CTD and the risk of pneumoconiosis)

There was a risk ratio nonreferent log relative risks $\log RR$ corresponding to doses (CTD) for each study (a total of six studies were included in this analysis), which are normally taken at the midpoint of each exposure group's range (just like the description in method section).

The data is then centralized, which is done by subtracting the values of the other central points in each study from the values of the smallest central points, which is defined as *CTDC*.

Following that, a two-stage random-effects dose-response meta-analysis was used to examine a possible non-linear relationship between CTD and the risk of pneumoconiosis. When a common 3-knot (10%, 50%, and 90% of the distribution) restricted cubic spline transformation is performed to the vector of aggregated exposure data (containing the reference category midpoint), a matrix of two spline transformations and two variables, *CTDCS1* and *CTDCS2*, are generated. First, we used a generalized least squares trend estimation method to produce study-specific estimates under a constrained cubic spline model. Second, we pooled study-specific results in a multivariate random-effects model using the restricted maximum likelihood method.

The first variable, *CTDCS1*, in the restricted cubic spline model, is defined as linear components according to the principle of building nonlinear components in restricted cubic spline models.

Furthermore, the nonlinear component of the restricted cubic spline model is the overall other nonlinear variables. Only one variable, *CTDCS2*, is a nonlinear component in this analysis. As a result, we determine if *CTDCS2* is statistically significant in this analysis by computing the p-value of the invalid null hypothesis test for *CTDCS2*, in which *CTDCS2* was assumed to be equal to 0. (a linear dose-response analysis). With the help of stata software calculation, the P value can be obtained as 0.16. Therefore, statistically, it is accepted that the *CTDCS2* is equal to 0, that is, there is a linear dose-response relationship between the variables.

Using the two-stage generalized least squares trend estimation method, a linear dose-response connection between an additional 1 mg/m³-years and the risk was studied. Same as the above procedure. First, Study-specific slope lines were computed, and then these lines were pooled to give an overall average slope.

Finally, using software, we calculated the regression coefficients of the risk ratio at the chosen reference level, *RRs* (relative risks for the spline models), and *RRI* (relative risks for the lines models) corresponding to the reference exposure level, as well as the 95% CIs (*lbs*, *ubs*, and *lbi*, *ubi*) of each level exposure level under the spline and linear models, respectively, and plot them. It is worth noting that we picked 50 mg/m³-years as the reference exposure level because this is the generally accepted amplitude of CTD alterations. As a result, rather than a nonlinear dose-response connection, there is a straight dose-response association between CTD and the risk of pneumoconiosis.

We sought to further quantify this link when the results revealed a linear dose-response relationship between CTD and the risk of pneumoconiosis. The initial outcome of the dose-response study, however, was a change in the risk of pneumoconiosis for every 1 mg/m³-year change in CTD, and a

change of 1 mg/m³-years was too tiny for CTD. For the reasons stated in the discussion section, we selected to calculate the change in risk of pneumoconiosis for every 50 mg/m³-year rise in CTD. According to our study and calculations, the risk of pneumoconiosis increases by 1.38 (95% CI: 1.22-1.55) with every 50 mg/m³-year increase in CTD. The forest plot in *Fig. 4d* depicted the change in pneumoconiosis risk for every 50 mg/m³-year rise in CTD in each study, as well as the overall results using this linear dose-response model.

Summary

We made the following changes to better clarify the relevance of the dose-response relationship between CTD and the risk of pneumoconiosis:

“An increase in duration by 5 years was associated with a 26% higher risk of pneumoconiosis (RR=1.26, 95%CI: 1.15-1.39, P<0.01; *Fig. 4c*) with heterogeneity (I²=97.68%, P<0.01), regarding 2.5 years as a reference.”, addressed on **page 18, lines 12-15**.

As well as the dose-response association between the durations and risk of pneumoconiosis, “a 50 mg/m³-years increase in CTD was associated with a 38% higher risk of pneumoconiosis (RR=1.38, 95%CI: 1.22-1.55, P<0.01; *Fig. 4d*) with heterogeneity (I²=99.75%, P<0.01), regarding 50 mg/m³-years as a reference.”, addressed on **page 19, lines 5-8**.

We included both the splines and the lines in the revised *Fig. 4* to help readers and reviewers understand the method of dose-response analysis, and the statistically significant line in the graph is black. And, in response to the reviewer's comments, we have added "*The detailed description of the dose-response analysis process*" in the *Supplementary Materials*.

To be honest, despite consulting with some professionals, mathematics, and stata software engineers during the dose-response analysis, we still did not fully understand the total process and principles due to a lack of knowledge in other fields. We carefully analyzed the data we collected in the same way that others else did (for example, *Naghshi S, Sadeghi O, Willett WC, Esmailzadeh A. Dietary intake of total, animal, and plant proteins and risk of all-cause, cardiovascular, and cancer mortality: a systematic review and dose-response meta-analysis of prospective cohort studies. BMJ. 2020 Jul 22;370:m2412. doi: 10.1136/bmj.m2412. PMID: 32699048; PMCID: PMC7374797*) So, I genuinely hope the reviewer will pardon us for not being the most thorough in the analysis process, but we did our best.

2. Response to comment: since the response variable is the incidence rate of silicosis, ordinary linear regression is not suitable even with the logarithm transformation. Generalized linear models are more appropriate.

Response: We learned a lot from this comment, and we completely agreed with the reviewer and referred to other similar studies (one of the studies: *Bose S, Rivera-Mariani F, Chen R, Williams D, Belli A, Aloe C, McCormack MC, Breysse PN, Hansel NN. Domestic exposure to endotoxin and respiratory morbidity in former smokers with COPD. Indoor Air. 2016 Oct;26(5):734-42. doi: 10.1111/ina.12264. Epub 2015 Dec 14. PMID: 26547489; PMCID: PMC5324735*). Finally, we deleted the linear regression model analysis results, leaving only Pearson's correlation analysis results, which are addressed on **page 19, lines 17-20**.

3. Response to comment: The writing is poor, necessary edit is needed.

Response: We carefully edited the entire work to eliminate grammatical problems. In addition, we sought writing aid and had the book reviewed by a native English speaker.

VERSION 2 – REVIEW

REVIEWER	Zhao, Kun Zhejiang University
REVIEW RETURNED	02-Dec-2022
GENERAL COMMENTS	accept for publication

REVIEWER	Cai, Jingheng Sun Yat Sen Univ, Statistics
REVIEW RETURNED	02-Dec-2022

GENERAL COMMENTS	My responses have been addressed.
-----------------------------------